

# Meteorological, snow and soil data, CO₂, water and energy fluxes, from a low-Arctic valley in the forest-tundra ecotone of Northern Quebec

Georg Lackner[1,2,3], Florent Domine[2,3,4], Denis Sarrazin[3], Daniel F. Nadeau[5,6], and Maria Belke-Brea[2,3,7]

[1]CNRM, Université de Toulouse, Météo-France, CNRS, Toulouse, France
[2]Takuvik Joint International Laboratory, Université Laval (Canada) and CNRS-INSU (France), Québec, Canada
[3]Centre d'Études Nordiques, Université Laval, Québec, Canada
[4]Department of Chemistry, Université Laval, Québec, Canada
[5]Department of Civil and Water Engineering, Université Laval, Québec, Canada
[6]CentrEau, Université Laval, Québec, Canada
[7]Presently at: Department of Biology, Wilfrid Laurier University, Waterloo, Canada

*Correspondence to*: Florent Domine (florent.domine@gmail.com)

**Abstract.** As the vegetation in the Arctic changes, tundra ecosystems along the southern border of the Arctic are becoming greener and gradually giving way to boreal ecosystems. This change is affecting local populations, wildlife, energy exchange processes between environmental compartments, and the carbon cycle. To understand the progression and the implications of this change in vegetation, satellite measurements and surface models can be employed. However, in situ observational data are required to validate these measurements and models. This paper presents observational data from two nearby sites in the forest–tundra ecotone in the Tasiapik Valley near Umiujaq in Northern Quebec, Canada. One site is on a mixture of lichen and shrub tundra. The associated data set comprises 9 years of meteorological, soil and snow data as well as 3 years of eddy covariance data. The other site, 850 m away, features vegetation consisting mostly of tall shrubs and black spruce. For that location, 6 years of meteorological, soil and snow data are available. In addition to the data from the automated stations, profiles for snow density and specific surface area were collected during field campaigns. The data are available at https://doi.org/10.1594/PANGAEA.946538 (Lackner et al., 2022b).



# 1 Introduction

The forest–tundra ecotone (FTE) marks a transition zone where the open-canopy forest of the boreal biome merges with the treeless Arctic tundra biome. According to Callaghan et al. (2002), the FTE spans more than 13 400 km across the northern parts of North America, Asia, and Europe, with a width of up to several hundred kilometers. This makes it the world's largest vegetation transition zone. A trend towards increased vegetation has been observed in the FTE. In fact, the above-ground biomass is on the rise for all Arctic environments (Meredith et al., 2019). Future projections indicate that the areal extent of tundra vegetation will decrease by at least 50% by 2050 (Pearson et al., 2013), while woody shrubs and trees will expand to 24–52% of the current tundra region, or 12–33% if tree dispersal is restricted (Meredith et al., 2019). This change will have major repercussions, such as large reductions in the soil carbon content due to more frequent wildfires (Mack et al., 2011) and widespread permafrost degradation occurring at increased rates compared to when only the changing environmental conditions are considered (Jones, 2015).

To understand, quantify, and project changes in the FTE, satellite monitoring and surface modeling are essential. However, both require in situ measurements. Although satellites cover large parts of the Earth's surface and are able to estimate a variety of surface-related variables (e.g., surface temperature (Qu et al., 2019), turbulent heat fluxes (Jiménez et al., 2017), vegetation cover (Guo et al., 2020)), most must be calibrated and/or validated using point measurements (e.g., Boisvert et al. (2015); Riihelä et al. (2017); Martin et al. (2019)). Surface schemes for climate models also require validation using in situ data (Krinner et al., 2018). Despite this need for data, few stations in Arctic regions are equipped to measure large sets of variables over long periods of time.

The Tasiapik Valley in northern Quebec, Canada is located within the FTE (Latifovic et al., 2017). This is an ideal location for conducting research because the lower valley is covered in open boreal forest while the upper valley consists of shrub and lichen tundra. Arctic and boreal biomes, as well as mixtures of both, are therefore present in close proximity to each other. Meteorological, snow, and soil data were collected starting in September 2012, and annual field surveys were conducted to study snow and soil characteristics (Domine et al., 2015). Turbulent heat fluxes were measured between 2017 and 2020 using the eddy covariance technique.

Detailed annual snowpit data are extremely valuable, as studies have shown that current snow models struggle to accurately simulate vertical profiles of density and thermal conductivity (Domine et al., 2016; Barrere et al., 2017; Gouttevin et al., 2018; Royer et al., 2021; Lackner et al., 2022a). Although new models are being developed to account for this deficiency (Jafari et al., 2020; Simson et al., 2021), critical validation data for snow density profiles remain very rare in the Arctic. In this paper, we present information on two research sites while fully documenting all the available data and providing a detailed analysis of the soil properties at the sites. We provide a comprehensive data set with meteorological, snow, soil, and turbulent flux data from 2012 to 2021.



## 2 Site Description

The study site is located in the Tasiapik Valley (Fig. 1) close to the village of Umiujaq, Quebec, Canada (56.55861°N,
76.48222°W). The valley forms a small catchment 4.5 km long and 1.3 km wide, and borders Tasiujaq Lake at an elevation of
0 m. The climate is subarctic with a mean annual temperature of −4.0°C. No long-term precipitation records exist, but our
recent data indicate a rather high mean annual precipitation compared to typical subarctic climates, at between 800 and 1000
mm. Around 50% of the precipitation occurs as snow. There is usually continuous snow cover from late October to early June.
Hudson Bay to the west of the valley (4 km distance) strongly influences the weather pattern. There is frequent fog throughout
the year (Robichaud and Mullock, 2001). Advection fog often forms in July and August when warmer air moves over the cold
Hudson Bay. The precipitation pattern is influenced by the extent of the ice cover in Hudson Bay. After freeze-up, the
precipitation rate drops and remains rather low until spring. Precipitation then increases in summer and peaks in late summer
and fall. The heat storage of Hudson Bay in summer and the subsequent release in fall also affects air temperatures, resulting
in relatively cold summer temperatures and warmer fall temperatures.

In Tasiapik Valley, vegetation is fairly spatially heterogeneous. In the upper valley, a mixture of lichen (*Cladonia* sp., mostly
*C. stellaris* and *C. rangiferina*), and shrubs with dwarf birch (*Betula glandulosa*), and other shrub species (*Vaccinium* sp.,
*Alnus viridis* subsp. *crispa* and *Salix* sp. including *S. planifolia*) with heights between 0.2 and 2 m dominate. Live lichen are
present not only on lichen tundra, but also in the understory of birches less than 80 cm tall. Live lichen can form in layers that
are 5 to 20 cm thick over a 2 to 4 cm layer of dead lichen, progressively transitioning to a thin organic litter layer (Gagnon et
al., 2019). The litter layer is only about 2 cm thick under lichen tundra and up to 5 cm thick under 80 cm tall birch. Taller
birches such as those found in water tracks have a mossy understory with a 10 cm thick organic layer (Gagnon et al., 2019).
Towards the bottom of the valley, vegetation turns into a forest-tundra with black spruce (*Picea mariana*) covering about 20%
of the surface. Below the open canopy, numerous birches are present. In areas not covered by woody vegetation, a variety of
grasses and mosses cover the surface.

There is discontinuous to sporadic permafrost in the valley (Lemieux et al., 2020) due to the presence of permafrost mounds
(lithalsas). At the exact location of the experimental setup, no permafrost was present. However, lithalsas were found within
30 m of the upper valley site. The soil composition is detailed in Sect. 7.

On 28 September 2012, a comprehensive meteorological station called TUNDRA (56.55877°N, 76.48234°W; elevation: 132
m) was deployed (Figure 1). Instruments were placed on a tripod. On 15 February 2013, snow temperature and thermal
conductivity sensors were installed on a vertical post a few m from the tripod, in dwarf birch 30 cm tall. Soil temperature and
humidity were measured starting on 19 September, 2015. One set of instruments was located under lichen, and one under low
birch near the post holding the snow sensors.

The FOREST station (56.55308°N, 76.47258°W, elevation: 82 m) was set up on 21 September 2015 with the same instruments
as those at the TUNDRA station. A fast-response gas analyzer with a sonic anemometer (model IRGASON, Campbell
Scientific, USA) was mounted at a 10 m tower about 15 m north of the TUNDRA station on 10 June 2017 and was operational

until 30 April 2020. A second set of temperature and thermal conductivity posts were installed on 20 September 2018 on lichen with no shrubs, about 15 m northwest of the tripod at TUNDRA. Multiple Reconyx time lapse cameras took several pictures per day and were installed in order to monitor the instrumentation and their surroundings. A complete list of all instruments deployed at TUNDRA and FOREST, as well as when each instrument was installed and at what precise position, is provided

in Tables 1 and 2, respectively. The data obtained are available in Lackner et al. (2022b).

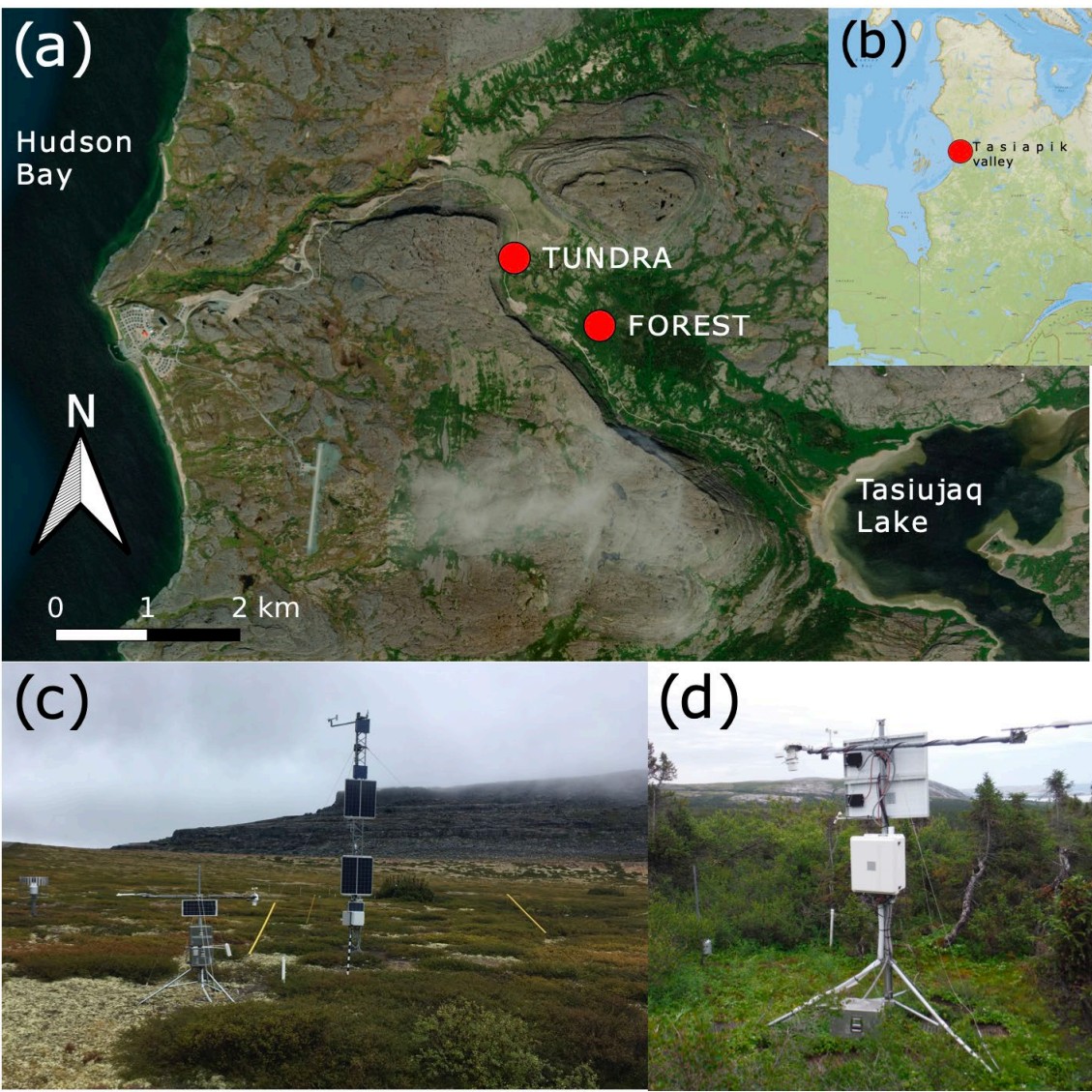

**Figure 1: (a) Location of the two sites in the Tasiapik Valley. (b) Location of the valley along the eastern shore of Hudson Bay in Northern Quebec. (c) Photo of the instrumentation at TUNDRA and (d) the instrumentation at FOREST. Source (a and b): ESRI.**





**3    Climate Data**

**3.1 Air Temperature, Humidity, Atmospheric Pressure, and Wind Speed**

   **3.1.1 TUNDRA**

Air temperature and relative humidity (RH) were measured at TUNDRA with a HC2-S3-XT sensor installed at a height of 2.3 m. No large data gaps were present. Small data gaps were filled using information from another sensor mounted close by on

the 10-m tower. As the RH measurements never reached ice saturation in winter, we corrected the raw values using linear equations based on air temperature in order to reach ice saturation. The method has been detailed in Domine et al. (2021).

**Table 1: Instrumentation at the TUNDRA study site.**

| Variable | Instrument | Manufacturer | Height/Depth | Years available | Comment |
|---|---|---|---|---|---|
| Air temperature | HC2-S3-XT sensor, inside white PVC tubing, ventilated | Rotronic | 2.3 m | 2012–2021 | |
| Atmospheric pressure | IRGASON | Campbell Scientific | 4.2 m | 2017–2020 | Before 2017 and after 2020 corrected ERA5 data |
| Wind speed | A100 Cup anemometer | Vector instruments | 2.3 m | 2012–2021 | |
| Radiation | CNR4 with CNF4 heater/ventilator | Kipp & Zonen | 2.3 m | 2012–2021 | |
| Precipitation | T-200 BM | Geonor | 2.3 m | 2016–2021 | Data before 2016 from uncorrected ERA5 |
| Snow depth | SR50 acoustic gauge | Campbell Scientific | | 2012–2021 | |
| Snow temperature | Pt1000 themistor | Hukseflux | 4, 14, 29, 44 and 64 cm | 2015–2021 | Near TUNDRA (SNOW1) |
| Snow temperature | Pt1000 and thermcouples | Hukseflux | 2, 2.5, 5, 7, 9.5, 12, 17, 21, 27, 32, 37, 42, 44.5, 47, 49.5, 52, 57, 62, 67, 72, 77, 82 cm | 2018–2021 | Near TUNDRA (SNOW2) |
| Snow thermal conductivity | TP08 | Hukseflux | 11, 21, 35, 53, 71 cm | 2015–2021 | Near TUNDRA (SNOW1) |
| Snow thermal | TP08 | Hukseflux | 7, 27, 47, 77 cm | 2018–2021 | Near TUNDRA |





| Variable | Instrument | Manufacturer | Height/Depth | Years available | Comment |
|---|---|---|---|---|---|
| conductivity | | | | | (SNOW2) |
| Soil temperature and volumetric water content | Decagon 5TM | Decagon (now METER) | 6, 12, 21, 32, 50 cm | 2015–2021 | Near TUNDRA under lichen |
| Soil temperature and volumetric | Decagon 5TM | Decagon (now METER) | 9, 15, 27, 39, 50 cm | 2015–2021 | Near TUNDRA under shrubs |
| Scenery | Time lapse camera | Reconyx | 1.5 m | 2015–2021 | Multiple cameras pointing different directions |

**Table 2: Instrumentation at the FOREST study site.**

| Variable | Instrument | Manufacturer | Height/Depth | Years available | Comment |
|---|---|---|---|---|---|
| Air temperature and humidity | HC2-S3-XT sensor, inside white PVC tubing, ventilated | Rotronic | 2.3 m | 2015–2021 | |
| Wind speed | Cup anemometer | Vector instruments | 2.3 m | 2015–2021 | |
| Radiation | CNR4 with CNF4 heater/ventilator | Kipp & Zonen | 2.3 m | 2015–2021 | |
| Snow depth | SR50 acoustic gauge | Campbell Scientific | | 2015–2021 | |
| Snow temperature | Pt1000 | Hukseflux | 4, 14, 29, 64 cm | 2015–2021 | Near FOREST (SNOW3) |
| Snow thermal conductivity | TP08 | Hukseflux | 4, 14, 29, 64 cm | 201–2021 | Near FOREST (SNOW3) |
| Soil temperature and volumetric water content | Decagon 5TM | Decagon (now METER) | 5, 10, 20, 30, 50 cm | 2015–2021 | Near FOREST |


From June 2017, measurements of specific humidity (SH) were collected with an IRGASON infrared gas analyzer. However, this instrument is susceptible to measurement errors caused by rain, snow, dew, or any other particles within the pathway of the gas analyzer. Complications with the IRGASON analyzer are detailed in Sect. 5. Gaps in the SH time series were filled





using converted RH measurements. The atmospheric pressure for SH was measured from June 2017 onward with the
IRGASON analyzer. Before that date and after dismantling the instrument in April 2020, ERA5 data were used. ERA5 is a
reanalysis product from the European Centre for Medium-Range Weather Forecast that provides hourly estimates for various
meteorological    and    soil    variables    starting    from    1959,    at    a    spatial    resolution    of    30    km
(https://www.ecmwf.int/en/forecasts/datasets/reanalysis-datasets/era5). However, as the ERA5 data do not correspond with the
same elevation, we corrected for a ≈10 hPa offset between the ERA5 data and the observations that was detected for times
when both sets of data were available. Except for two long power outages (see Sect. 5), there were no other significant gaps in
the time series for pressure. The gaps from the power outages were filled using the corrected ERA5 data.

Wind speed data were collected with a cup anemometer at a height of 2.3 m at TUNDRA. In winter, the instrument was
sometimes  stalled  due  to  frost  during  stable,  low-wind  conditions.  To  fill  those  gaps,  we  used  data  from  a  Young
anemometer affixed at a height of 10 m on the nearby tower (CEN,1997–2020). At times, the Young anemometer became
covered in ice at the same time as the cup anemometer. During those times, we used the data from the FOREST station, where
the instrument was installed at the same height as at TUNDRA. During one period in January 2021, all available instruments in
the valley were stalled. We therefore used an instrument from the UMIROCA station (CEN, 1997-2020), a station located
on  the  shore  of  Hudson  Bay.  All  data  used  to  fill  the  gaps  were  corrected  using  a  linear  regression.  This  was  done  to
account for different installation heights and environments (vegetation, elevations, topography). Figure 2 shows the time series
of all the variables mentioned above.



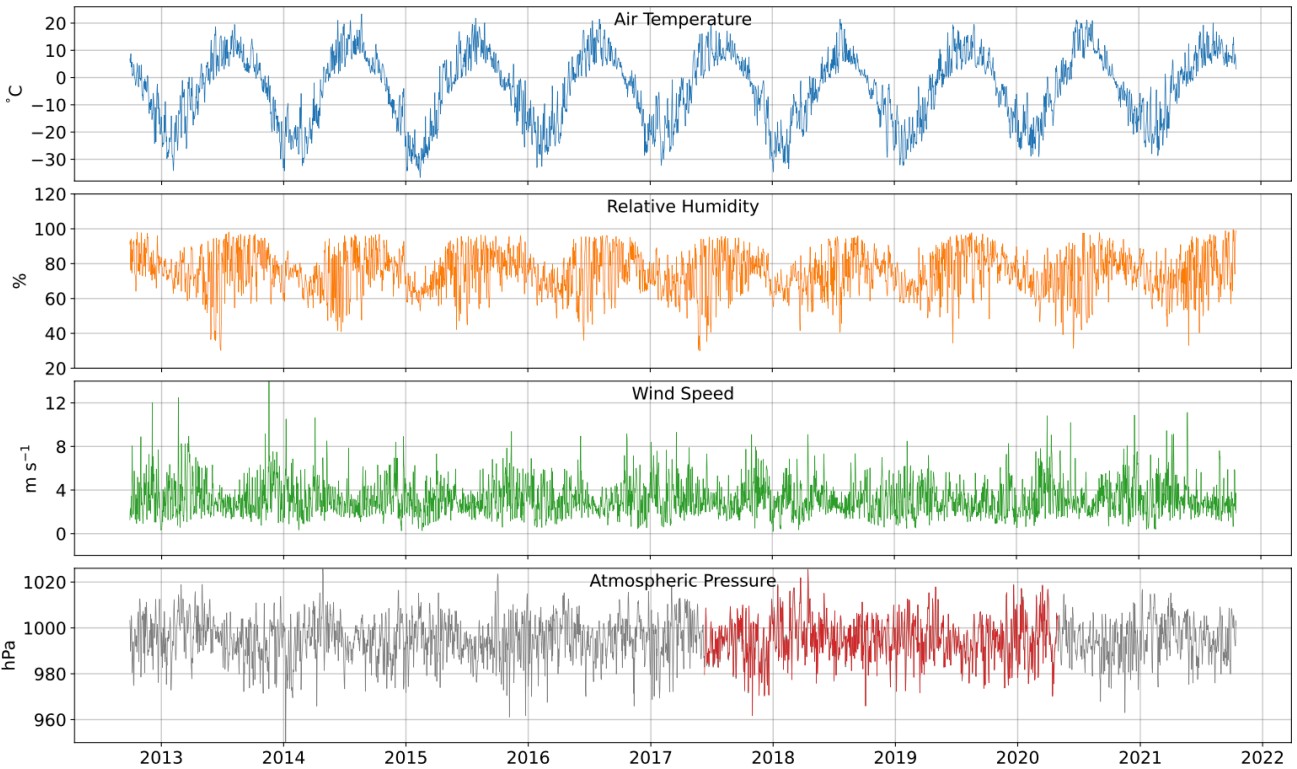

**Figure 2: Time series of hourly air temperature, relative humidity, wind speed, and atmospheric pressure at TUNDRA. The red section of the atmospheric pressure curve represents local observations, while the grey sections represent ERA5 reanalysis data.**


### 3.1.2 FOREST

At the FOREST site, air temperature and wind speed were recorded with the same instruments as at TUNDRA (see Fig. 3). The temperature time series had no large gaps (> 3 h), and small gaps (≤ 3 h) were filled by interpolation. We used values from TUNDRA to gap fill data for FOREST when the cup anemometer was ice-covered in winter, and applied a linear regression to

adjust wind speeds. There was also an RH sensor at FOREST, but due to malfunctions, the recorded data could not be used.

Compared to TUNDRA, temperatures at FOREST were slightly higher throughout the year, except in November and December. From January 2016 to December 2020, the mean difference was ≈1°C. Since the surface roughness is greater at FOREST than at TUNDRA, wind speeds were lower for specific heights. Indeed, the mean wind speed from January 2016 to December 2020 was about ≈1 m s$^{-1}$ lower at FOREST than at TUNDRA.



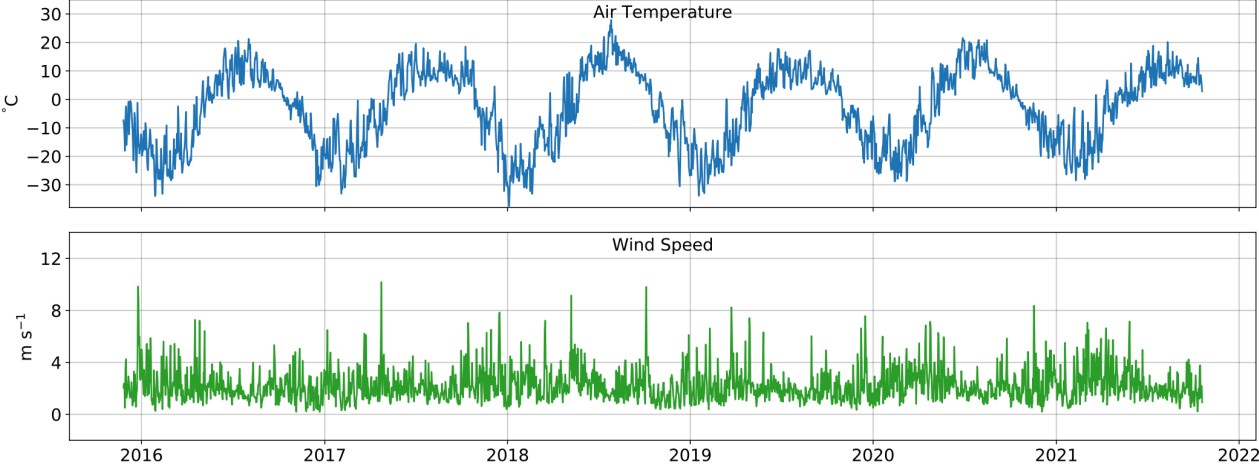


**Figure 3: Time series of hourly air temperature and wind speed at FOREST.**

## 3.2 Radiation

### 3.2.1 TUNDRA

The surface radiation terms were measured using a 4-component radiometer (model CNR4, Kipp & Zonen, The Netherlands) mounted at 2.3 m above ground. It was equipped with a CNF4 heating/ventilation unit (Kipp & Zonen, The Netherlands), which mostly prevented snow accumulation and the build-up of frost and dew on the measuring lenses. The CNF4 was programmed to be active for 5 min every hour, just before radiation measurements were collected.

From 10 October 2018 to 27 September 2020, no temperatures were recorded in the CNR4, which were otherwise used to
correct the longwave radiation. During this period, we estimated CNR4 temperatures using a gradient boosting regressor from scikit-learn (Pedregosa et al., 2011) with the following input variables: air temperature, radiation, humidity, and wind speed. We trained the decision tree using data from periods when the CNR4 temperatures were recorded and found a good agreement between the observed and estimated values. Using the estimated temperatures, we calculated the longwave radiation ($LW$) using:

$$LW = (5.67 \times 10^{-8}) \frac{V}{C} T^4, \tag{1}$$

where $V$ is the measured output voltage, $C$ is the calibration constant and $T$ is the temperature of the instrument in $K$.

In October 2018, the entire CNR4 unit was replaced and recalibrated. As the calibration constants changed, we applied a correction to account for the drift. We assumed that the calibration constants varied linearly over time between both
calibrations. Between October 2018 and September 2021, no recalibration took place. We therefore used the constants of the



instrument deployed in October 2018 without time variations.

Small, mostly negative values were observed at night for up- and downwelling shortwave radiation, although both these values are typically expected to be zero. To compensate for this discrepancy, we calculated the mean offset for both components and subtracted them from the respective upwelling and downwelling radiation.

Despite the CNF4 heating unit, the accumulation of frost and snow sometimes interfered with the incident radiation measurements (long and shortwave). The exact periods when the CNR4 sensors were impacted by frost and snow could only be determined visually. Since a detailed visual inspection could only be performed at the site, we applied several quality control criteria to the downwelling radiation (both long and shortwave) to exclude the affected periods. Therefore, all values at times when the wind speed was < 0.5 m s$^{-1}$ or the uncorrected longwave downwelling radiation was > −5 W m$^{-2}$ were discarded if

the air temperature was < 0°C. Subsequently, gaps of up to 3 hours were interpolated, while longer gaps were filled using corrected ERA5 data. The correlation between ERA5 data and observations was established using the remaining data that passed our quality control measures. The 9-year time series for the 4 radiation components at TUNDRA are shown in Fig. 4.

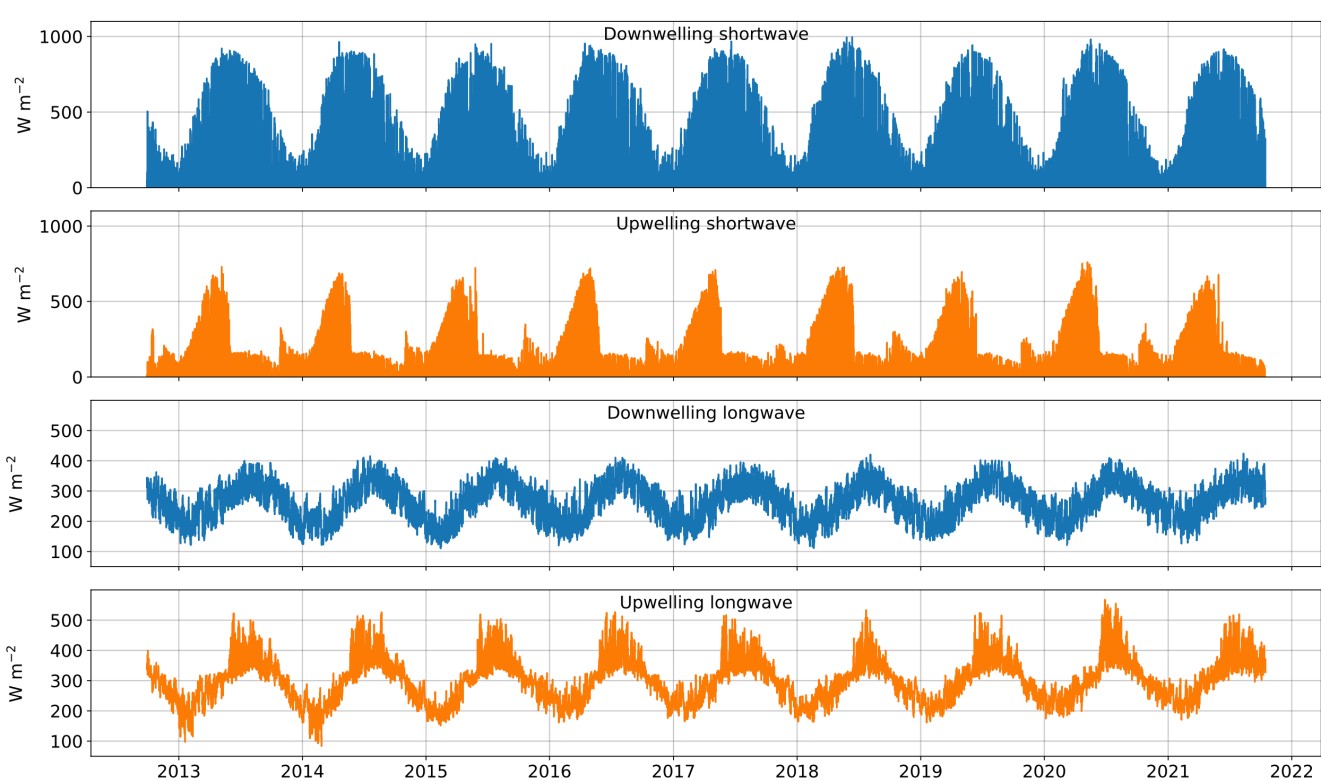

**Figure 4: Time series of hourly downwelling and upwelling shortwave and longwave radiation at TUNDRA.**


### 3.2.2 FOREST

The FOREST station setup was similar to that at TUNDRA, with a CNR4 radiometer at a height of 2.3 m combined with a CNF4 heating/ventilation unit. The CNR4 at FOREST was recalibrated in October 2021. The raw values were corrected for the

drift of the calibration constants. No power outages or instrumental failures occurred at this site and the complete time series from 28 September 2015 is shown in Fig. 5.

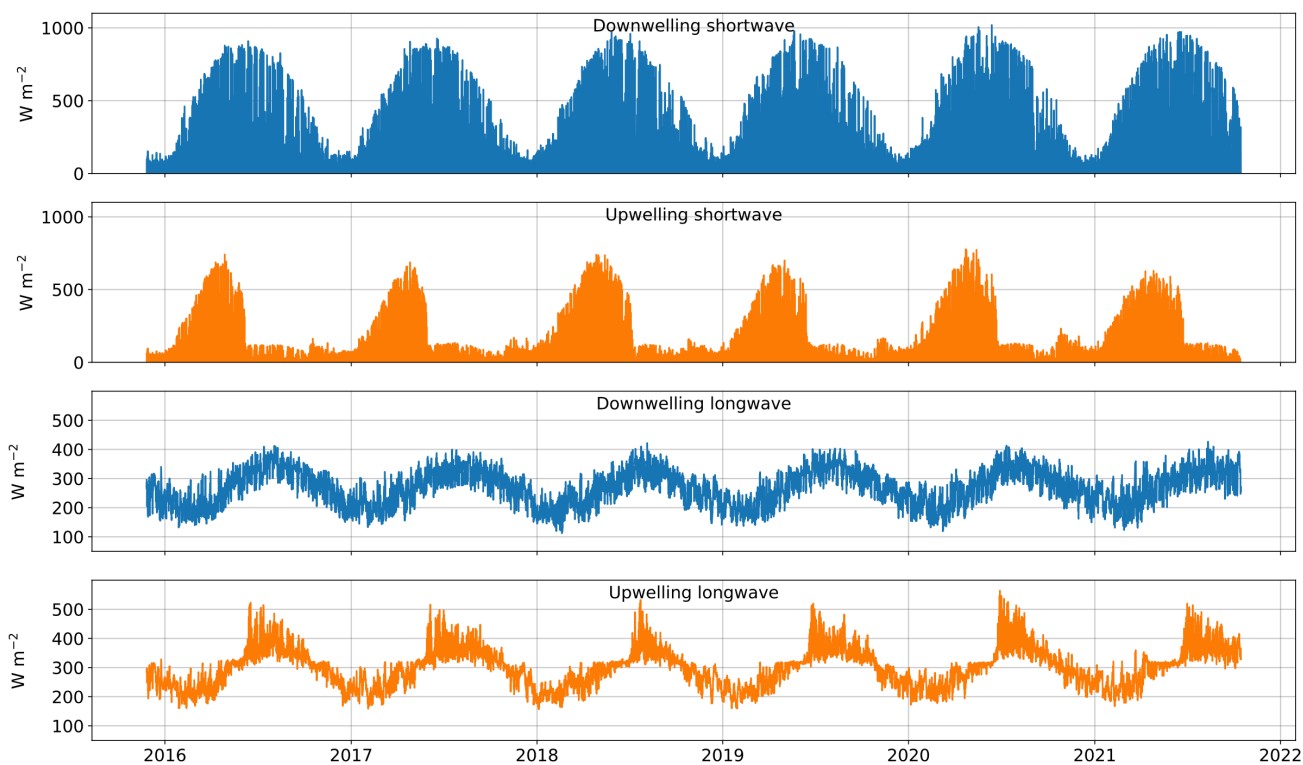

**Figure 5: Time series of hourly downwelling and upwelling shortwave and longwave radiation at FOREST.**

At FOREST, we also observed small, non-zero values at night for the shortwave radiation. To account for these offsets, we

applied the same procedure as for TUNDRA. The same problems with frost and snow build-up were observed at FOREST. We applied the same criteria as for TUNDRA. The linear equations for the ERA5 data are also shown in the Supplementary Material.

Small differences became apparent when we compared the radiation observations at both sites. The downwelling shortwave radiation was smaller at FOREST, which can be attributed to greater topographic shading. Between January 2016 and December

2020, when data from both stations were available, the mean difference was 4.15 W m$^{-2}$. For the longwave counterpart, there was very little difference in summer, but a small deviation was detected in winter when the longwave downwelling radiation was slightly higher at FOREST. This might be an effect of the higher vegetation levels at FOREST. Radiation from the steep ridges surrounding the valley may also contribute to the longwave downwelling radiation. However, for the same 4-year period,

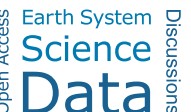

the difference was only 1.5 W m$^{-2}$. Differences in upwelling radiation were slightly higher (TUNDRA−FOREST: 4.6 W m$^{-2}$
for shortwave radiation and −2.15 W m$^{-2}$ for longwave radiation). However, these values heavily depend on the radiative and
thermal properties of the surface and soil, as well as on the duration of the snow-free period.

### 3.3 Precipitation

In May 2016, a T200B precipitation gauge (Geonor, USA) equipped with a single Alter shield was installed to measure solid
and liquid precipitation close to the TUNDRA station. The gauge recorded hourly cumulative precipitation ($PR_{tot}$) in kg m$^{-2}$
h$^{-1}$, equivalent to mm h$^{-1}$. The standard deviation ($\sigma_i$) of each hourly measurement was also recorded. The gauge has an inlet
with a diameter of 16 cm and the rain/snow is collected in a cylinder with a capacity of 1000 mm. An anti-freeze agent was
added in the cylinder to melt snow and keep the stored water from freezing. The use of an anti-freeze agent is preferable to
a heating system, as heat increases water loss due to evaporation, particularly in summer. Evaporation is further reduced by
adding a thin layer of oil to the water surface. Three vibrating wire load sensors weigh the entire cylinder and provide three
independent measures for mass. First, the raw cumulative values from the three vibrating wire load sensors were transformed
into hourly mass variations. Occasional erratic fluctuations occurred, induced by perturbations of the wire load sensors by the
wind and other factors. Data that were obviously inconsistent given the latitude, for instance those beyond $\pm$ 30 mm h$^{-1}$, were
eliminated and the three independent precipitation rates ($PR_i$ with $i$ = 1, 2, 3) were combined using a weighted mean. Each
hour, the wire load sensor with the highest standard deviation ($\sigma_i$) was removed and the weighted mean was computed using
the remaining two values, with the inverse of the standard deviation defining the weights as $w_i = 1/\sigma_i$, such that

$$PR_{tot} = \frac{w_1 PR_1 + w_2 PR_2}{w_1 + w_2}. \tag{2}$$

Subsequently, the precipitation was partitioned into snow and rain using a single threshold of 0.5°C, determined through visual
observations at the airport weather station, located only 3.2 km away. In addition, a correction for the underestimation of solid
precipitation in the presence of wind (undercatch) was applied following Kochendorfer et al. (2018),

$$PR_{cor} = PR_{uncor} \frac{1}{0.742 \exp(-0.181U + 0.332)} \tag{3}$$

where $PR_{cor}$ is the corrected precipitation rate in mm, $PR_{uncor}$ is the uncorrected precipitation (in mm), and $U$ (in m s$^{-1}$) is the
wind speed at the height of the gauge orifice, provided by the nearby weather station. Prior to installing the precipitation gauge
with the single alter shield and three independent wire load sensors, a simpler version with a home-made alter shield and
only one wire load sensor was present at the site. However, this setup did not produce reasonable data and therefore the data
were discarded. For the period between 2012 and 2016, we used only ERA5 data, as it proved to be closer to our observations

than data from the two closest meteorological stations from Environment and Climate Change Canada (Kuujjuarapik, 160 km south and Inukjuak, 230 km north). When comparing summer and winter monthly precipitation observations with ERA5 data, we observed no biases for the summer values. However, we detected an underestimation of ERA5 for the winter months, with cumulative precipitation exceeding 50 mm. To correct for this bias, the values for November to April were multiplied by 1.3822 for months with a cumulative precipitation greater than 50 mm. Otherwise, no correction was applied to the ERA5 precipitation

data.

In the lower panel in Fig. 6, the daily precipitation at TUNDRA is shown from 2016 to 2021, as well as the ERA5 precipitation data before 2016. The upper panel in Fig. 6 depicts the seasonal cumulative precipitation for each summer and winter period, respectively. The dates associated with the onset of snow and meltout are shown in Table 3, and were determined using snow gauge data and time lapse cameras.

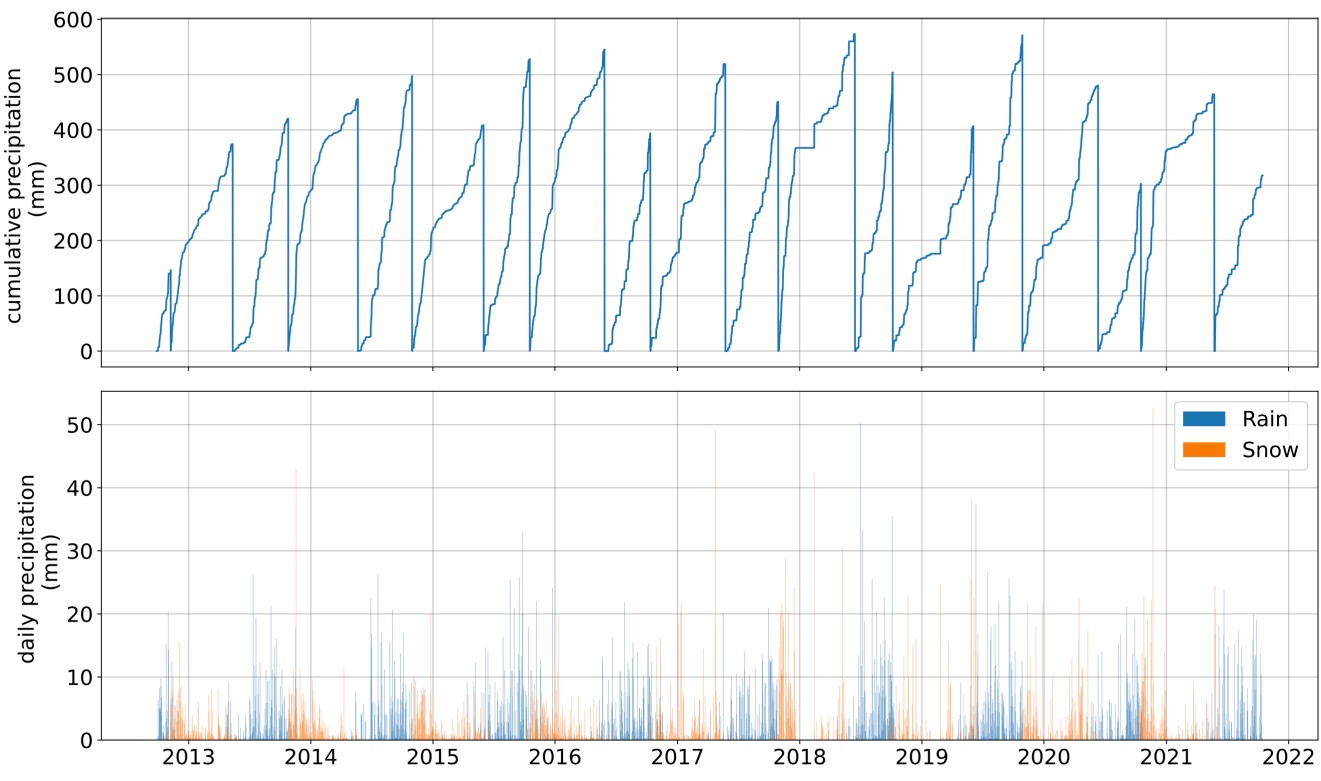


**Figure 6: Time series of cumulative precipitation and daily rain and snow. Before 28 May 2016, ERA5 data were used.**




**Table 3: Snow onset and meltout dates at the TUNDRA site, used to determine seasonal cumulative precipitation.**

| Snow year | Snow onset | Meltout |
|---|---|---|
| 2012–13 | 9 November 2012 | 31 May 2013 |
| 2013–14 | 26 October 2013 | 22 May 2014 |
| 2014–15 | 31 October 2014 | 28 May 2015 |
| 2015–16 | 18 October 2015 | 28 May 2016 |
| 2016–17 | 12 October 2016 | 24 May 2017 |
| 2017–18 | 29 October 2017 | 15 June 2018 |
| 2018–19 | 6 October 2018 | 4 June 2019 |
| 2019–20 | 29 October 2019 | 11 June 2020 |
| 2020–21 | 17 October 2020 | 24 May 2021 |

## 4. Spectral and Broadband Albedo

The surface albedo for various types of vegetation cover was measured between 12 and 18 September 2015 for wavelengths between 346.5 and 2513 nm with a portable field spectrometer (HR-1024 portable spectroradiometer, Spectra Vista Corporation). The radiation signal over this spectral range was monitored with a Si photodiode (346.5 to 982 nm) and two InGaAs photodiodes over the 982–1882 and 1882–2513 nm ranges. For wavelengths greater than 2340 nm, upwelling irradiance was very low, resulting in a mostly unusable signal. We therefore present only the results for the 346.5–2340 nm

range. The radiation signal was collected by an integrating sphere placed at the end of a 3 m rod to minimize interference from the person taking the measurements. The horizontal position of the sphere was ensured by an electronic inclinometer next to the sphere. The downwelling signal was collected first. Then, the sphere was rotated 180° to record the upwelling signal. A photodiode monitored the solar radiation to ensure that it remained constant (within 1%) during both measurements. Spectra were smoothed over 10 nm intervals for wavelengths shorter than 1780 nm, and over 60 nm intervals for longer wavelengths.

Five spectra were recorded over areas with lichen cover in the vicinity of the TUNDRA site. Five spectra were also recorded over short birch shrubs with lichen understories in the same area. We visually estimated that > 90% of the leaves were still green. The FOREST site consisted of a mixture of spruce that reached up to 3 m high, birch and grass. As such, it was not possible to obtain a representative spectrum of the entire FOREST area, as this would have required measurements from a height of at least 10 m. We therefore measured eight spectra of dense, short spruce within a few km of the FOREST station (Fig. 7).

Lastly, we measured the spectra of grassy surfaces with little to no erect vegetation and with little to no lichen. Although these spectra were not necessarily recorded at the FOREST site, the grassy vegetation was fairly similar at both locations. The average spectra for all four types of vegetation are shown in Fig. 7. The broadband (BB) albedo (346.5–2340 nm) of each spectrum was calculated from the ratio of the integrated upwelling radiation to downwelling radiation. The average BB albedos were 0.203 for lichen, 0.155 for birch, 0.174 for spruce and 0.180 for low grassy vegetation.





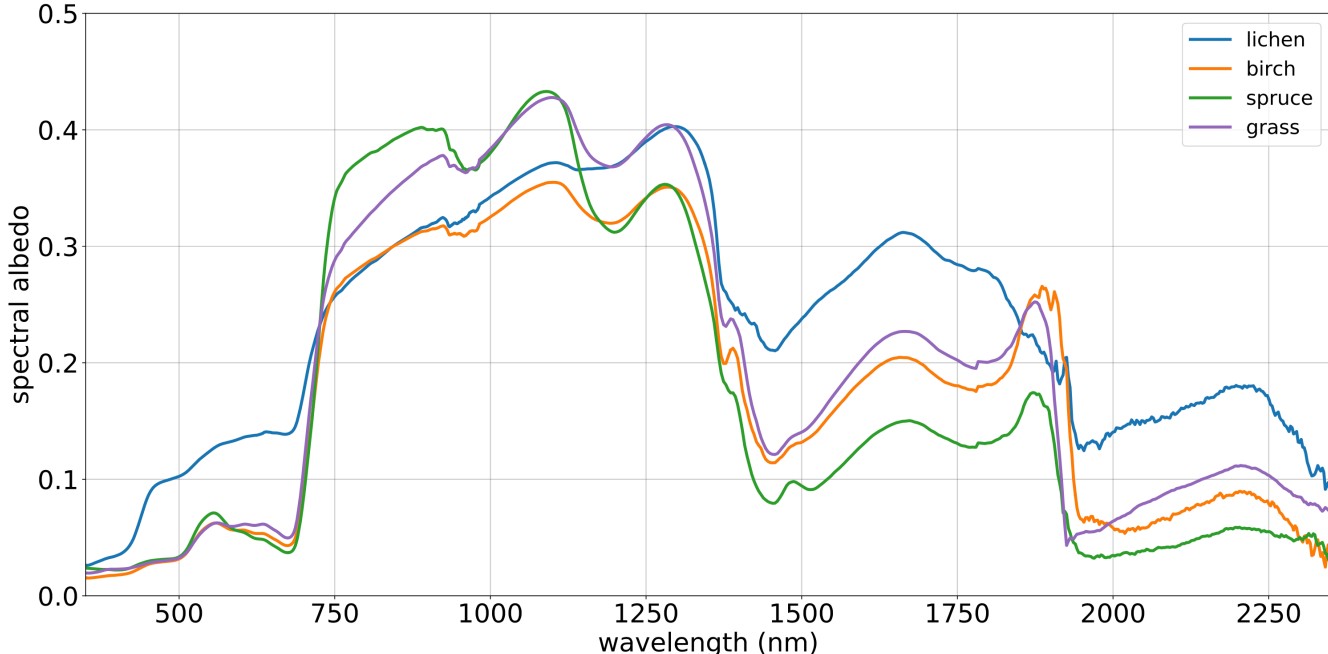

**Figure 7: Average spectral albedo for the four types of vegetation cover.**

Variations in the spectral albedo were observed for the measurements at the different sites, as detailed in Table 4. Variations between sites were smallest for lichen, increased for birch and spruce, and were highest for low grassy vegetation, Variations in birch and spruce are probably mostly due to differences in the leaf area index and in the amount of woody vegetation present. Differences in low grassy vegetation are due to variations in species and the occasional presence of short shrubs, such as *Vaccinuim sp.* and *Betula glandulosa*.

**Table 4: Variation in albedo within each vegetation class, at 550 and 1100 nm.**

|  | 550 nm | 550 nm % variation | 1100 nm | 1100 nm % variation |
|---|---|---|---|---|
| Lichen | 0.116 to 0.131 | 11.5% | 0.359 to 0.393 | 8.6% |
| Low birch | 0.051 to 0.067 | 23.9% | 0.305 to 0.387 | 21.2% |
| Spruce | 0.058 to 0.107 | 45.8% | 0.355 to 0.559 | 36.5% |
| Low grassy | 0.053 to 0.080 | 33.7% | 0.370 to 0.523 | 29.3% |

These data allowed for the estimation of the BB albedo of the FOREST site. We estimated that the vegetation coverage is 25% spruce, 40% low grassy vegetation, and 35% birch, leading to a BB albedo of 0.170. We estimated the TUNDRA site to be 60% lichen and 40% birch, with a BB albedo of 0.184.



## 5 Turbulent Flux Data

Turbulent heat fluxes were measured at TUNDRA using a fast-response sonic anemometer and a $CO_2$/$H_2O$ infrared gas analyzer (IRGASON, Campbell Scientific, USA) installed 4.2 m above ground on the 10-m tower. The three components for wind speed and concentrations of $H_2O$ and $CO_2$ were recorded with a CR3000 datalogger (Campbell Scientific) at a frequency of 10 Hz. The 10-Hz data were processed with the EddyPro® (version 7.0.3; Li-COR Biosciences, USA) software package. This software calculates 30-min averages of the turbulent heat and carbon fluxes and a set of corrections. These corrections are

the detrending of turbulent fluctuations based on a running mean, covariance maximization, density fluctuation compensation (Webb et al., 1980), and analytic correction of high-pass and low-pass filtering effects (Moncrieff et al., 1997). To align the coordinate system with the surface, we have chosen to apply a double rotation. The planar fit method from Wilczak et al. (2001) was also tested, but it was unsuccessful due to the presence of snow. In order to assess data quality, a random uncertainty quantification was used following Finkelstein and Sims (2001), which identified outliers, spikes, and artifacts. Finally, the 0-

1-2 quality scheme from Mauder et al. (2013) was applied to flag the data, and segments that were flagged as 2 (poor quality) were removed from the data set.

   To sort out the remaining outliers and to fill the gaps according to the EddyPro® procedure, post-processing was necessary. This was done using the PyFluxPro program (Isaac et al., 2017), which comprises six processing levels and uses EddyPro output files as inputs and produces a continuous time series for all fluxes. For the first three processing levels, data were read,

quality controlled, and finally, auxiliary measurements were merged when gaps were present. The quality control includes (i) range checks based on user-defined limits, (ii) spike detection, (iii) manual removal for specific dates, and (iv) data rejection based on other variables. Erroneous flux data were rejected based on $CO_2$ and $H_2O$ signal strengths from the infrared gas analyzer (IRGA) and internal error codes from both the sonic anemometer and the IRGA. For the fourth processing level, meteorological variables were gap-filled with ERA5 data. Each variable was bias-corrected using a linear fit between ERA5

and flux tower observations during periods when both were available.

   Finally, the fluxes were gap-filled using interpolation and a self-organizing linear output map (SOLO) – a type of artificial neural network (ANN) (see Hsu et al. (2002) and Abramowitz (2005)). Interpolation was only applied for gaps of up to 3 hours, while SOLO was used for longer gaps. SOLO requires a set of environmental drivers such as air temperature, radiation, and humidity, as well as the fluxes themselves as inputs. SOLO first constructs relationships between the environmental drivers by

applying an ANN-equivalent of a principal component analysis. It then uses an ANN-equivalent of a multiple linear regression to make connections between the drivers and the fluxes. ANN together with marginal distribution sampling (MDS, Reichstein et al. (2005)) was shown to be the best choice for gap-filling flux data (Moffat et al., 2007). The resulting series for the sensible and latent heat fluxes, as well as the $CO_2$ fluxes, are shown in Fig. 8.

   Since the IRGASON is an open-path sensor that is sensitive to external disturbances such as precipitation particles, gaps are

frequently present in the data set for the turbulent and $CO_2$ fluxes. The fraction of gaps subsequently increased with each processing step in EddyPro® and PyFluxPro. Overall, 27% of the sensible heat flux, 43% of the latent heat flux, and 44% of





the $CO_2$ flux data were gap-filled using the SOLO. These values include two longer outages of IRGASON in March 2018 and from January to March 2019. The interpolated data and data gap-filled with SOLO are specifically flagged.

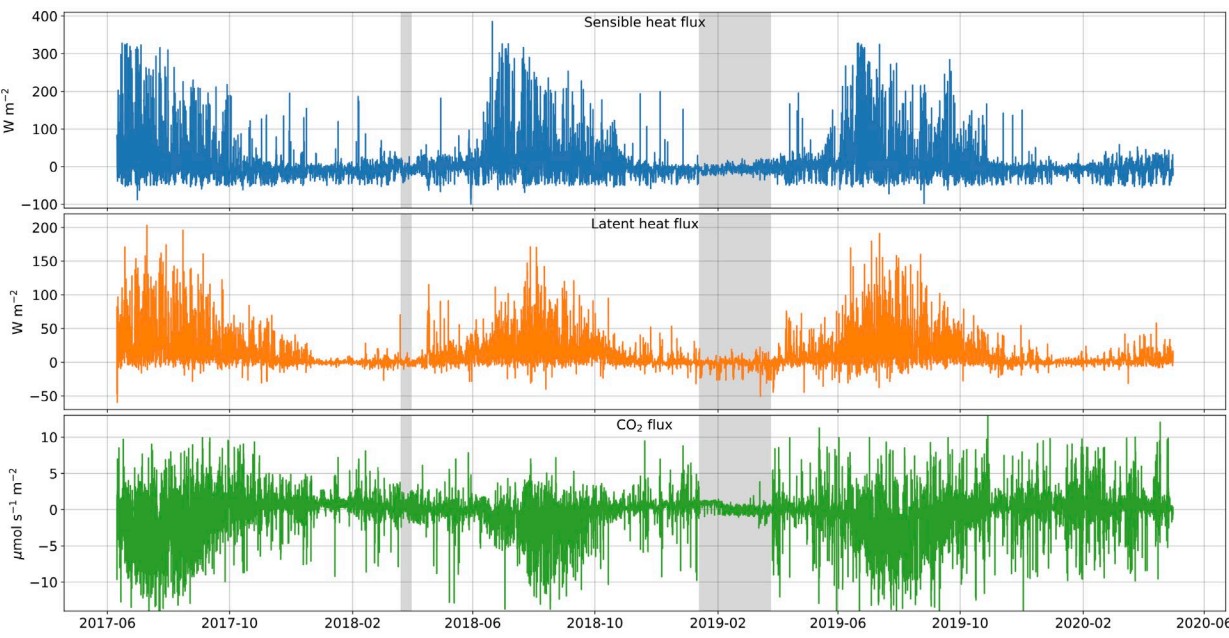

**Figure 8: Time series of hourly sensible and latent heat fluxes and the $CO_2$ flux. The grey shaded areas indicate power outages at the station during which no flux data were recorded.**

# 6 Snow Data

## 6.1 Snow Height

Two SR50 sonic distance sensors provided continuous snow height measurements near TUNDRA. One was installed exactly at the TUNDRA site and the other was mounted on the nearby 10-m tower. A snow height value of zero was assigned for the snow-free period in summer. Unfortunately, the snow height data were incomplete. We therefore decided to merge both data sets. The gaps that remained despite the merge were filled with estimates from time-lapse images of the snow poles. A similar SR50 sonic sensor was installed at FOREST. The snow height data at FOREST for winters 2016–17 and 2017–18 were also incomplete and thus, no data are shown for these periods. In spring, the snow height at FOREST almost reached the sensor. We observed during our field visits that the wind formed a depression on the snow surface just below the sensor. We thus estimate that snow height was underestimated by about 20% by the sensor in late March–early April. The time series for both stations are depicted in Fig. 9. Snow height values at FOREST are consistently larger than at TUNDRA due to the presence of taller vegetation, which more effectively traps blowing snow.





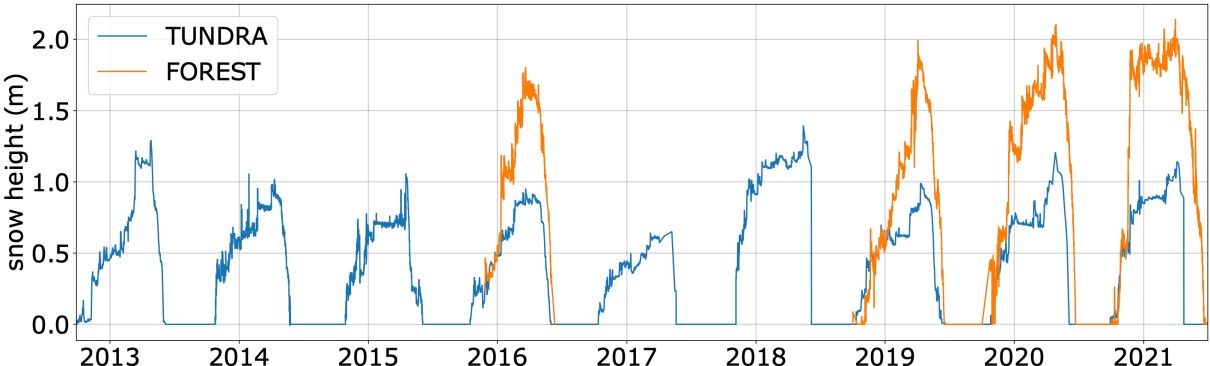

**Figure 9: Evolution of snow height from the automatic gauges at TUNDRA and FOREST.**

## 6.2 Snow Temperature

### 6.2.1 TUNDRA

Vertical profiles of snow temperature were recorded by two snow poles located approximately 4 m (SNOW1) and 15 m

(SNOW2) from TUNDRA. They were equipped with Pt1000 thermistors (which are part of the TP08 needles) and collected

temperature measurements every two days. SNOW1 was set up in 2015 in a patch of shrubs about 30 cm tall with a lichen

understory. The Pt1000 thermistors were installed at 4, 14, 29, 44 and 64 cm above the lichen. However, because lichen height

is somewhat arbitrarily determined, and given that the lichen might also be compacted by the snow, the indicated height data

contain some uncertainty. SNOW2 was installed in 2018 on a patch of lichen. Four Pt1000 thermistors were placed at 7, 27,

47, and 67 cm above ground. Again, there is some uncertainty associated with these heights, as the there is no consistent line

between the soil and lichen, making the zero reference slightly arbitrary. SNOW1 data are shown in Fig. 10.

For both stations, data associated with a positive snow temperature were deleted as they implied that the sensor was not buried

in the snow or that the sensor was heated by the sun through a thin snow layer. Using time-lapse images, we were able to

identify times when the thermocouples/thermistors were not covered with snow. However, as the camera was 10 to 15 m away

from the stations, we cannot rule out that some data from times with no snow were included. Note that different snow heights

and internal snow properties were observed at the two stations, and as such, the respective snow temperatures do not necessarily

match for a similar measurement level.





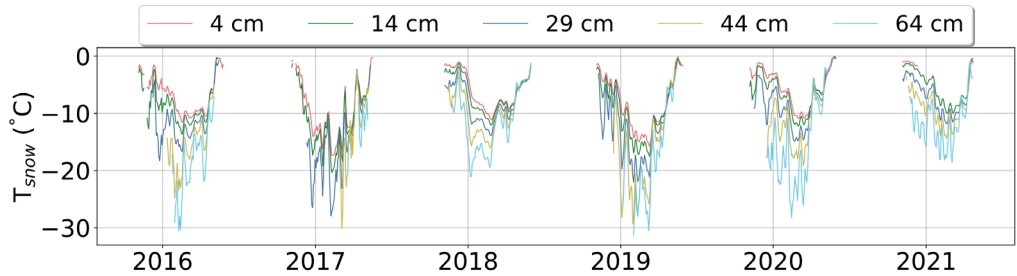

**Figure 10: Bi-daily time series of the Pt1000 thermistors from the SNOW1 station at TUNDRA at heights of 4, 14, 29, 44 and 64 cm.**

### 6.2.3 FOREST

At FOREST, another snow pole (SNOW3) was installed and equipped with four Pt1000 thermistors (at 4, 14, 29, and 64 cm). SNOW3 was placed in a patch of grass and moss. The heights associated with SNOW3 are reliable within 2 cm. No time-lapse camera was available at FOREST, and as seen in Fig. 9, the snow height time series was less complete. Thus, only positive temperatures were removed and no further data cleaning was performed. Figure 11 shows the snow temperature at the four measurement levels. Snow temperatures were substantially higher at FOREST due to the deeper snowpack.

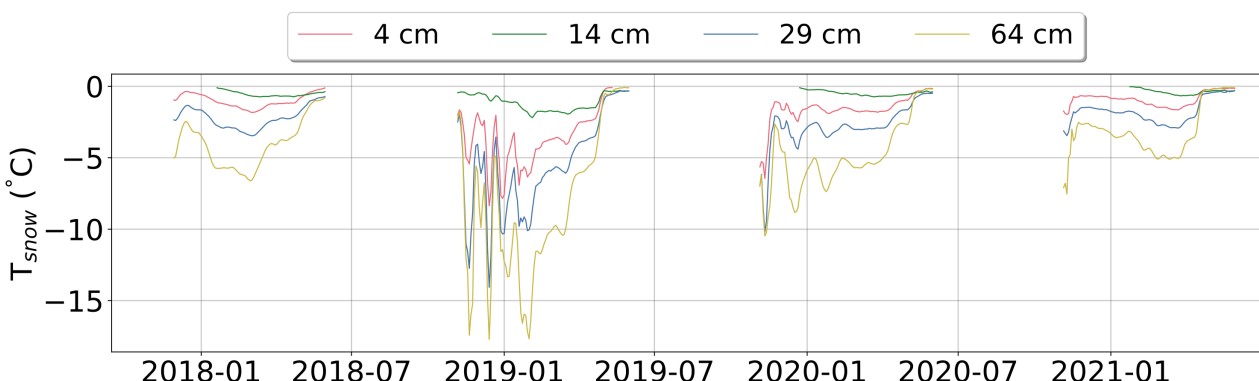

**Figure 11: Hourly time series of the Pt1000 thermistors from the SNOW3 station at heights of 4, 14, 29, and 64 cm.**

### 6.3 Snow Thermal Conductivity

#### 6.3.1 TUNDRA

TP08 heated needle probes were installed along with the temperature probes at both SNOW1 and SNOW2. The installation heights were the same as those for the Pt1000 thermistors. A description of the method used to determine the snow effective




thermal conductivity from the TP08 heated needle probes is provided in Domine et al. (2015). Figure 12 shows the observations from SNOW1 at five heights over 6 years.

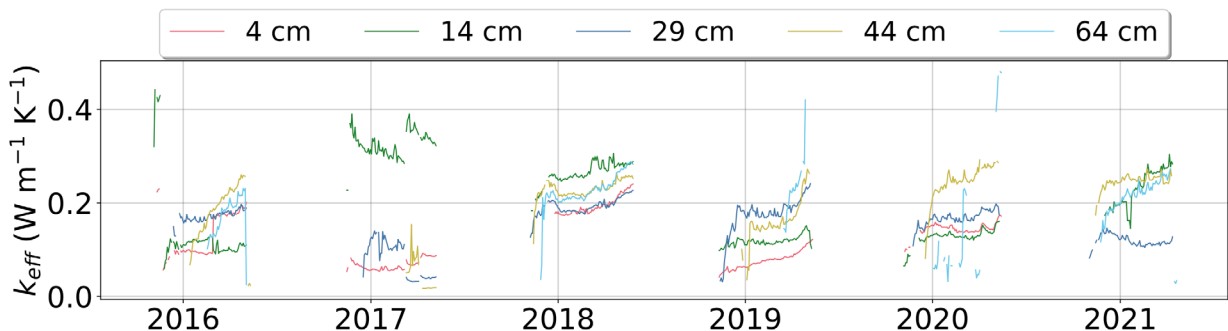

**Figure 12: Time series of the snow thermal conductivity from the SNOW1 station at heights of 4, 14, 29, 44 and 64 cm.**

### 6.3.2 FOREST

The thermal conductivity at FOREST was also recorded with the TP08 heated needles at SNOW3. These were also installed at the same heights as the Pt1000 temperature sensors. Because the sensors at 4 and 14 cm did not work properly, the corresponding values were not included. The recorded values at 29 and 64 cm are shown in Fig. 13.

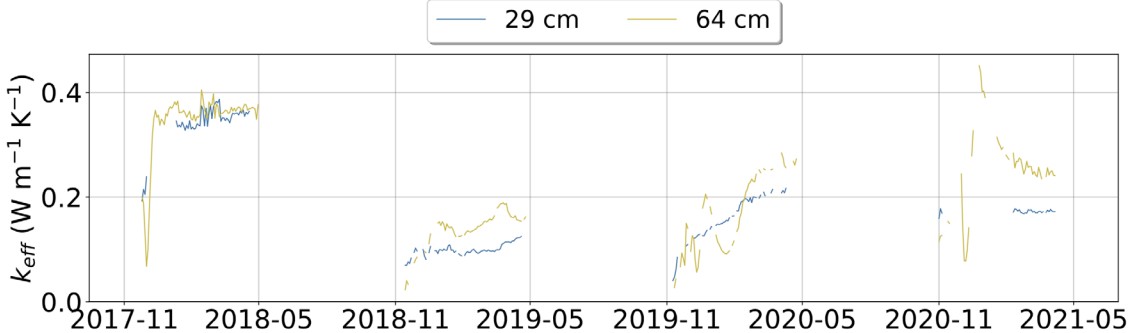

**Figure 13: Time series of the snow thermal conductivity from the SNOW3 station at heights of 29 and 64 cm.**

### 6.4 Manual Snow Measurements

Field trips were conducted for most years to measure snow density and snow specific surface area (SSA). Snow density was measured with a 100 cm$^3$ box cutter (Conger and McClung, 2009) and a field scale, while SSA was measured using infrared

reflectance in an integrating sphere as described in Gallet et al. (2009). In 2014 and 2016, we visited the site in January, while in 2013, 2014 and 2015, we conducted field trips in February. The site was visited in late March in 2015, 2016, 2017 and 2019,

while in 2018, it was visited in early April. The available snow data therefore provide an overview of the snow properties in mid-winter and early spring. Visits in 2020 and 2021 were impossible because of the COVID pandemic. The profiles of snow density and SSA are illustrated in Fig. 14. Note that the representativity of the measured profiles is limited, as the physical snow properties are highly spatially variable due to vegetation, micro-topography, and wind erosion and redeposition.

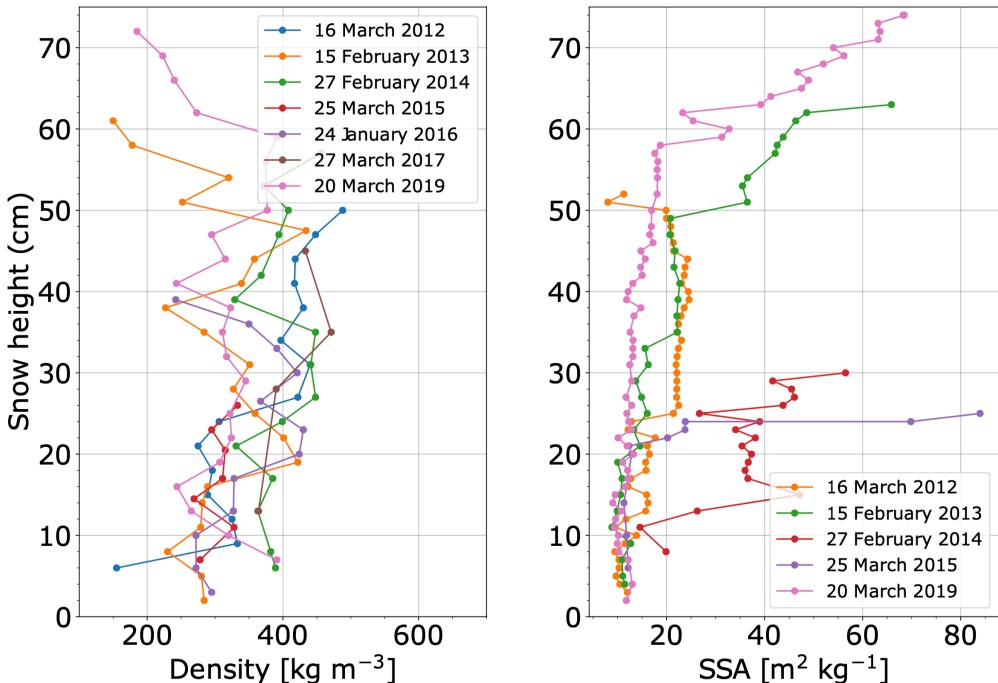

**Figure 14: Vertical profiles of snow density and SSA measured in the vicinity of the TUNDRA station from 2012 to 2019.**

## 7. Soil Data

### 7.1 Soil Temperature and Moisture

Soil temperatures at TUNDRA and FOREST were recorded using 5TM soil temperature and water content sensors. According to 5TM specifications, the resolution is 0.1°C for the soil temperature and 0.0008 $m^3$ $m^{-3}$ for the soil water content. The accuracy is 1°C for the temperature and 0.03 $m^3$ $m^{-3}$ for the soil water content. At all the stations, we observed offsets during the zero-curtain period, when $T = 0°C$. The temperatures were corrected for these offsets, ranging between 0.2°C and 0.6°C.

#### 7.1.1 TUNDRA

The soil temperature and soil water content were measured at two sites near TUNDRA, each with a different type of vegetation. Multiple Decagon 5TM probes were used for these measurements. Figure 15 shows these values for a lichen-covered surface, while Fig. 16 shows values for low-shrub vegetation. The soil temperature at the lichen-covered site is warmer during the





summer months compared to the low-shrub site, and the soil water content is generally lower. This might be due to more shading from the shrubs and the differences in soil composition, as detailed in Sect. 7.2.

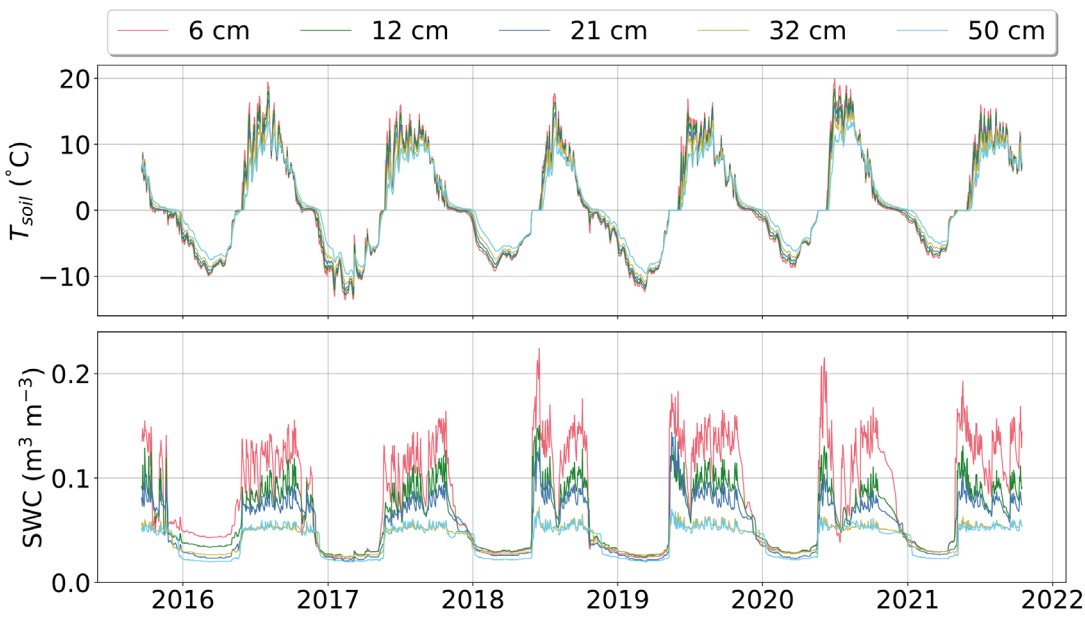


**Figure 15: Time series of the daily soil temperature and the soil water content (SWC) under a lichen-covered surface at depths of 6 cm, 12 cm, 21 cm, 39 cm, and 50 cm at TUNDRA.**

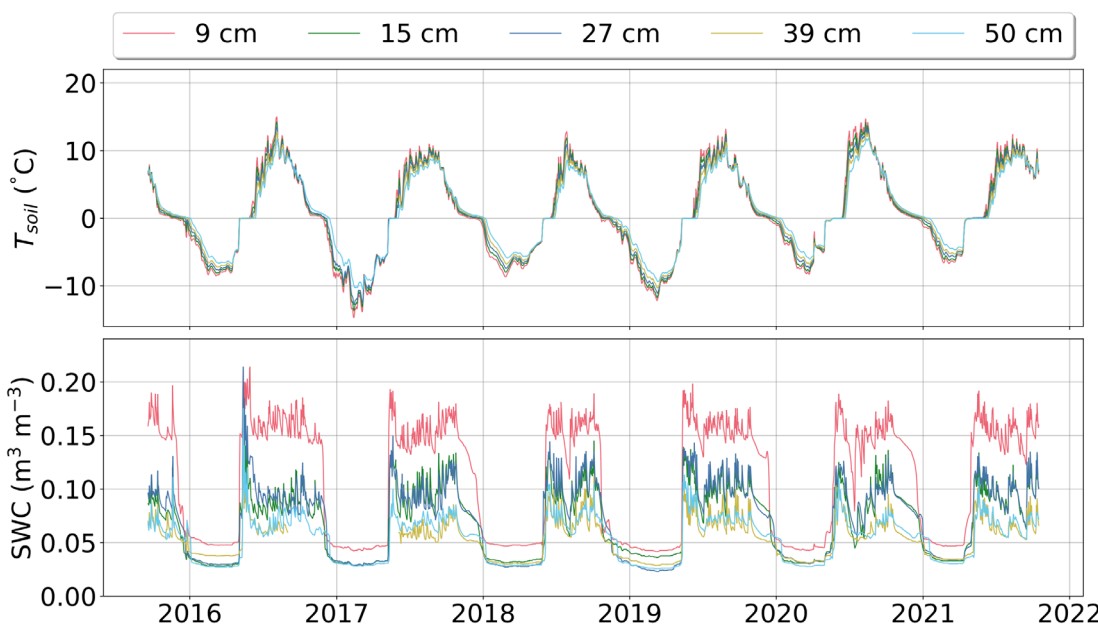

**Figure 16: Time series of the daily soil temperature and the soil water content (SWC) volume under a patch of low shrubs at depths of 9 cm,  15 cm, 27 cm, 39 cm, and 50 cm at TUNDRA.**





### 7.1.2 FOREST

The soil temperature and water content were measured at FOREST using the same instruments as the TUNDRA site (Fig. 17).

The sensors were placed about 80 cm from the SNOW3 pole. The soil water content and temperatures at FOREST were very

distinct from those measured at the TUNDRA site. During summer, the soil temperatures were slightly cooler than those under

the low-shrub surface. However, in winter, the soil freezes late and the minimum temperatures were only slightly below 0°C

due to the thick snow cover. The soil water content is substantially higher at FOREST than at TUNDRA because the soil

contains less sand compared to TUNDRA and because more snow accumulates in winter at FOREST and melts in spring.

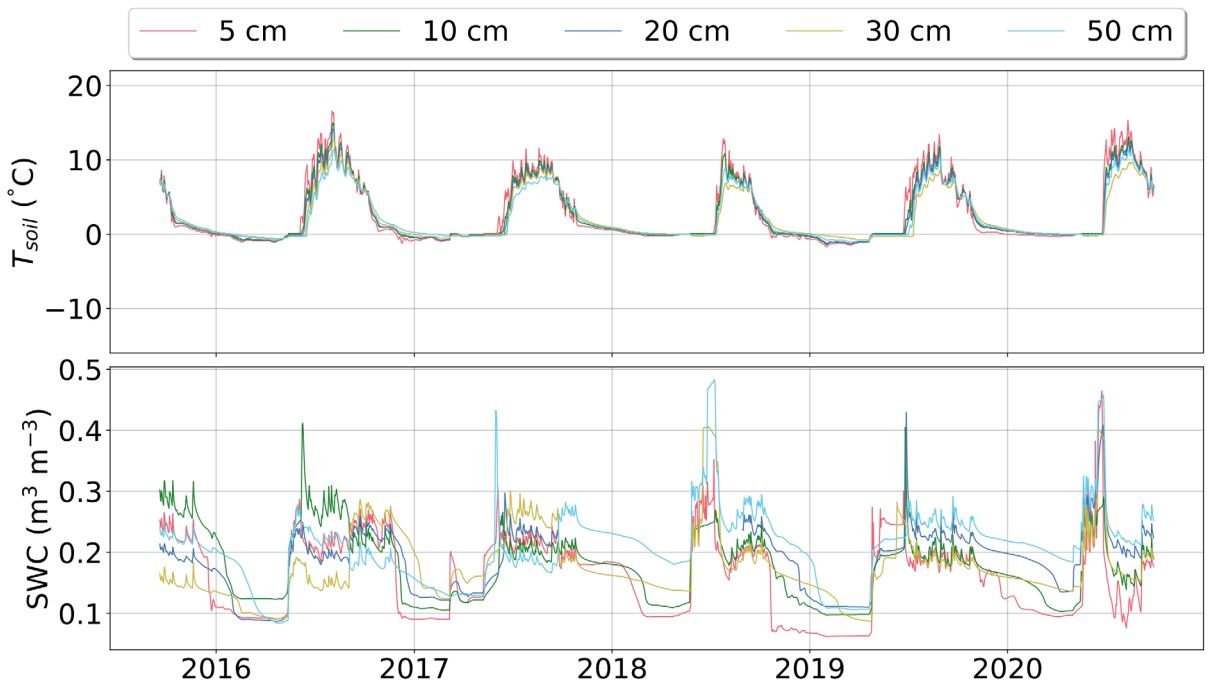

**Figure 17: Time series of the daily soil temperature and the soil volume water content (SWC) below grass and moss at depths of 5 cm, 10 cm, 20 cm, 30 cm and 50 cm at FOREST.**

### 7.2 Soil Properties

The Tasiapik Valley consists of former beaches that have been uplifted by isostatic rebound after the Laurentide Ice Sheet

melted a few millennia ago. According to Bhiry et al. (2011), land at an elevation of around 130 m, such as the TUNDRA site,

emerged 6500 to 7000 years ago. The FOREST site, at an elevation 82 m, emerged about 5000 years ago. Because they were

formerly beaches, the soil at both TUNDRA and FOREST sites is mostly sandy. Gagnon et al. (2019) conducted granulometric

analyses at TUNDRA and reported a unimodal particle size distribution of around 500 μm (pure sand), with an occasional,

small, secondary peak at around 80 μm (loamy sand). Based on two soil pits, we estimate the sand fraction of the soil at

FOREST to be lower than at TUNDRA, but no granulometric analyses were performed there.

The organic carbon content of the soil at TUNDRA is among the lowest in the Arctic and subarctic (Gagnon et al., 2019), with about 1.5 kg m$^{-2}$ of organic C in lichen tundra, and 4.2 kg m$^{-2}$ in low birch shrubs (< 80 cm). No detailed soil analyses were performed at FOREST, but two soil pits were dug and revealed an organic litter layer 6 to 10 cm thick. The organic carbon content of the soil at FOREST was not measured, but given the thick litter layer, it is probably greater than at TUNDRA.

Soil thermal conductivity and density profiles are shown in Fig. 18. On 7 and 8 October 2014, short-distance spatial variation
tests were performed, revealing changes within a range of 25% over a horizontal distance of 30 cm, and at a depth of 20 cm. Overall, there was a clear trend for an increase in thermal conductivity with depth. The thermal conductivity of lichen was also measured and it was essentially the same as that of air, forming an efficient insulating layer in summer. In winter, snow crystals blend into the lichen. Therefore, we determined that assuming only a depth hoar snow layer while ignoring the lichen is likely adequate. We recommend using thermal conductivities of 0.12 W m$^{-1}$ K$^{-1}$ for the top 5 cm of the soil (starting at the base of
the live lichen), 0.5 W m$^{-1}$ K$^{-1}$ for depths between 5 and 10 cm, 0.9 W m$^{-1}$ K$^{-1}$ between 10 and 20 cm, and 1.1 W m$^{-1}$ K$^{-1}$ for depths below 20 cm. We only measured three density profiles, which showed an increase in density down to 10 cm in depth, and then remained at an almost constant value of around 1500 kg m$^{-3}$. Between depths of 0 and 10 cm, the density is approximately of 1000 kg m$^{-3}$. Sand has a specific heat of about 796 J kg$^{-1}$ K$^{-1}$ (Carvill, 1993).

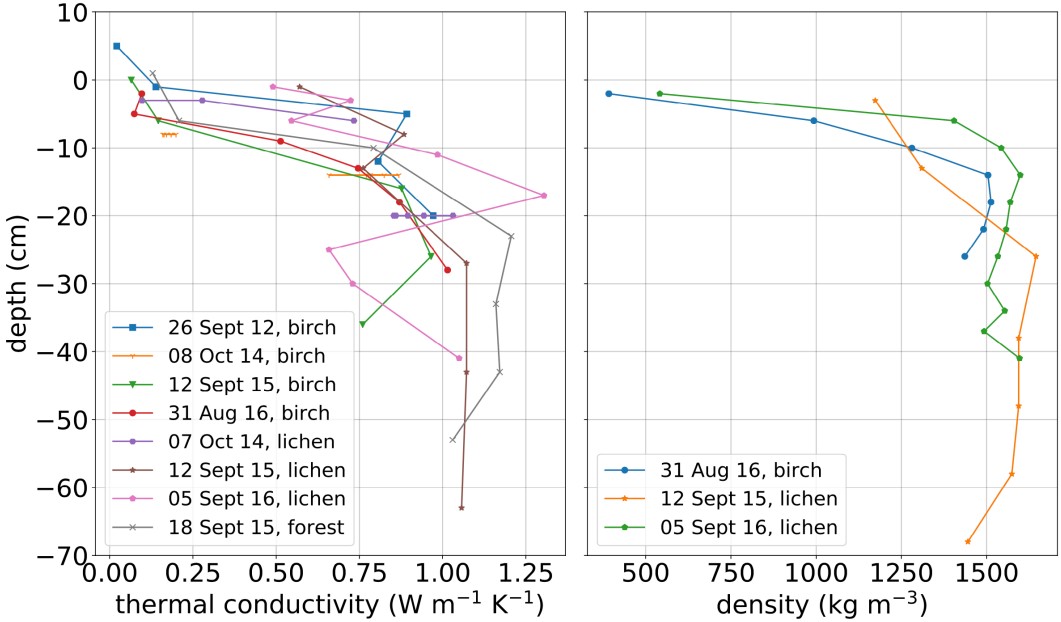

**Figure 18: Thermal properties of soils. (a) Thermal conductivity; (b) density. Depths were measured from the base of the live lichen.**

**8. Conclusions**

The increasing temperatures in Arctic regions are causing substantial environmental changes, such as the thawing of permafrost

and the greening of the Arctic landscapes. Both effects are more pronounced along the southern border of the Arctic, where
the land is transitioning into a boreal forest. In this study, we present two data sets, a 9-year data set for TUNDRA and
a 6-year data set for FOREST that include numerous measurements in soil, snow and above the ground at two sites along the
treeline in eastern Canada. These data provide information that can be used to calibrate and improve earth system models,
particularly snow schemes, which showed poor performance when simulating Arctic snowpack properties. Our data can help
advance the understanding of the relationships between potential meteorological drivers, permafrost degradation and Arctic
greening.

**Acknowledgements.**

This work was funded by the French Polar Institute (IPEV), the Natural Sciences and Engineering Research Council of Canada (Discovery
Grant and Northern Research supplement programs), the BNP-Paribas foundation (APT project), and the European Union's Horizon 2020
research and innovation program under grant agreement No. 101003536 (ESM2025 – Earth System Models for the Future) and grant
agreement No. 727890 (INTAROS). We thank the community of Umiujaq for welcoming us and allowing this research to be conducted. We
benefited from facilities from the Centre d'Études Nordiques during the entire duration of this study. Compute Canada assisted with data
storage and handling. This work is a contribution to the IASC project Terrestrial Multidisciplinary distributed Observatories for the Study of
Arctic Connections (T-MOSAiC).

**Data availability**

The data are available on the PANGAEA repository, https://doi.org/10.1594/PANGAEA.946538 (Lackner et al., 2022b).

[For review, a perhaps easier data access is https://www.dropbox.com/sh/4j828hj4h2oee68/AACZS3yqD-_35r1K5_30B-S2a?dl=0]

**Author contributions.**

FD, GL and DN designed the research. FD and DN obtained funding. DS, FD, GL, and DN deployed and maintained the instruments. GL and
FD analyzed the data and prepared the data files. FD, GL, and MBB performed the fieldwork. GL and FD wrote the paper with comments from
MBB, DS and DN.

**Competing interests**

The authors declare that they have no conflict of interest.




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
