# Peer review of "Meteorological, snow and soil data, CO2, water and energy fluxes, from a low-Arctic valley of Northern Quebec"

_Earth System Science Data, 2023_

## Author Response (AR1)

Dear Editor,

Please find below our response to the Referee's comments and a detailed description of the changes we have made. As pointed out by Referee#2, modifications were required in the data files and we have therefore made a new submission to PANGAEA since this repository does not usually allow changes to published data sets.  PANGAEA informed us on 7 December that our "data submission has been initially checked and was approved for the next steps in our editorial workflow." They however added that "we are currently facing a high rate of data submissions to PANGAEA and thus the editorial process and minting of DOI names might take up to 12 weeks." We are therefore submitting this revised version with the data files accessible by a link, but they are not posted on PANGAEA yet. We hope that this will nevertheless allow the evaluation of this revised version.

The original first author, Dr. Lackner, has not been available since the reviews were received, and we have changed the authors' order in this revised version and in the new data set submitted to PANGAEA. We now detail our revision. We hope this revised version will be judged suitable for publication in ESSD, and we of course remain available to address any further comment you may have. We gratefully thank the Referees for their comments and their time, and in particular Dr Julie Friddell for very carefully checking the data files.

Sincerely,

Florent Domine, on behalf of all coauthors.

**Response to Referee #1.**

*Our responses, in blue italics, are embedded in the Referee's comments. We first thank the Referee for these useful and constructive comments.*

The paper describes data from two adjacent sites in the forest-tundra ecotone in Northern Quebec.

With one station in the tundra landscape and one in the forest this data set covers an important transition zone.

The data will be of great interest for validation and improving of (climate) models.

*Thank you for this positive evaluation.*

**Specific Comments**

**Abstract**

I would like to read a sentence here about your gap-filling practices

*Thank you for this suggestion. We do not think however that discussing gap-filling is useful in the abstract, especially that this procedure involves few data and would be lengthy to discuss in an informative manner in an abstract. All required information is given in the main text. We will listen to the Editor's recommendation on this point.*

**2. Site Description**

line 80: The presence of lithalsas are not the reason for the permafrost, but the indicator why we know/think there is permafrost. Please rephrase

*Thank you for this remark. We will change the text to "There is discontinuous to sporadic permafrost in the valley (Lemieux et al., 2020) as witnessed by the presence of permafrost mounds" lines 79-80.*

**3. Climate Data**

Table 1: could you make the rows more distinguishable? Shade every second row or something?

*We apologize for the lack of clarity. Shading is not allowed by the journal, but we will improve the layout. In any case, the final layout will be done at typesetting.*

I suggest to use negative numbers for soil depth to make it more clear which numbers are height over surface and which are depth below surface

*Thank you for this suggestion. This has now been done in Table 1.*

line 212: This sentence suggest you would allow precipitations value between -30 and 0 mm/h? There are now negative precipitation values in the data.

*We apologize for this error. Indeed, there cannot be negative precipitation values. We will change ± 30 mm h$^{-1}$ to 30 mm h$^{-1}$, line 218.*

**7. Soil Data**

Figure 15/16: could you make the accuracy of the tic marks and the grid the same for these two images? The resolution of your tic marks and the white space and the axis limits are often not the same for the pair of TUNDRA/FOREST images.

*We will modify Figure 15 so that it matches Figure 16. Due to the swapping of sections 7.1 and 7.2 (see subsequent Reviewer's comment), these will be Figs 16 and 17.*

line 422: Why did you choose to describe the soil properties after the measurements of temperature and soil moisture? I would suggest to change that, as it makes the interpretation of the soil moisture plots more intuitive

*Good idea. Done. This has not been tracked-changed however, to avoid cluttering the tracked-changes version.*

**Technical Comments**

line 24: "The data is available" instead of "The data are available"

*Rigorously, "data" is a plural word, with "datum" the singular. Many people use "data" as a singular word, but this unfortunately widespread use is not grammatically correct.*

line 345: delete "the" in ".. with these heights, as the there is.."

*Done, thank you.*

**Response to Referee #2**

*We first thank the Referee for these useful and constructive comments. In particular, we are grateful to Julie Friddell for carefully checking the data files.*

The data in this publication provide easy-to-access multi-year, high-resolution in situ records of environmental data from remote locations in the southern Arctic forest-tundra ecotone.  These data are difficult to obtain and, as validation and calibration inputs for satellite and modelling studies of climate-related changes in the far north, may serve as important contributors to future understanding of ecological changes in this dynamic transitional region.  Missing data are generally filled in with replacement data from nearby comparable instruments, regression on the adjacent data, or other model outputs (such as ERA5), so that uninterrupted time series are available.  Quality flags are applied to some variables.

The data publication is comprehensive, to introduce, explain, and visually represent this complicated and extensive data set, and provides sufficient references for further reading.  The data set is valuable, apparently unique, and useful.

*Thank you for this positive evaluation.*

**Detailed edits/questions to the manuscript:**

Lines 104-105:  Air temp and relative humidity - missing data (small gaps) provided by "another sensor" nearby at 10 m height - which instrument?  Please describe/give details?

*That other sensor is also a HC2-S3 Rotronic sensor. It is on the 10 m tower about 10 m from the tripod where our main sensor is located. This is visible in Figure 1c. This has now been specified in the text, lines 106-107.*

Table 1 - Should "Air temperature" also include "and humidity" as in Table 2?  "themistor" needs an "r" and "thermcouples" needs an "o".  Second "Soil temperature and volumetric" - should it have "water content" added to the end, like the one above?

*Thank you for spotting these errors. Yes, the Rotronic sensors also measures humidity. This will be corrected, along with the other errors.*

Table 2 - what are the dates for Snow Thermal Conductivity (the table currently says "201–2021")?

*Thanks again for spotting this. We meant 2015. We have also added thermal conductivity and snow temperature data for the TUNDRA site for the 2/2013 to 5/2015 period, and extra lines have been added to the Table to mention those.*

Lines 272-274: "Variations between sites were smallest for lichen, increased for birch and spruce, and were highest for low grassy vegetation, Variations in birch and spruce are probably mostly due to differences in the leaf area index and in the amount of woody vegetation present." This is two sentences and should have a period instead of a comma at "vegetation, Variations"

*Thank you, changed, line 281.*

Also, if I understand it correctly, Table 4 suggests that the variation is largest for spruce, not for grass, as is stated. Please review and revise as needed.

*Thank you for the comment. We checked the data and indeed the data from Table 4 are correct. We therefore changed the text to state that variations were highest for spruce, lines 279-281.*

Lines 291-292: "To align the coordinate system with the surface, we have chosen to apply a double rotation." Please provide a reference or brief explanation, as the procedure is not clear, as written.

*Rotating the coordinate system is a key step in the eddy covariance technique. The idea is to be able to correct the sensor's imperfect alignment with the surface and define a coordinate system whose z-axis is normal to the surface. As suggested, we have added a reference and a brief additional explanation (lines 299-301).*

*"To align the coordinate system with the surface, we have chosen to apply a double rotation (Wilczak et al, 2001). In brief, for each 30 min period, we perform two rotations to align the coordinate system with the flow streamlines, imposing zero lateral and vertical wind speed over the period."*

Figure 8 - Instead of 4 months, could the x-axis be given in 6 month periods, or seasonal (3 months), so that it is easier to visualize the annual pattern?

*Thank you for the suggestion. We will add 3-month intervals as suggested. Here is the new version of Figure 8.*

[Figure]

*Figure 8: Time series of hourly sensible and latent heat fluxes and the $CO_2$ flux. The grey shaded areas indicate power outages at the station during which no flux data were recorded.*

Line 345:  Remove "the" before "...there is no consistent line..."

*We have reworded this section (lines 354-355).*

Figure 11 - Please check the colours of the legend, as the red line is always between the green and blue lines, which would indicate that the 14 cm depth is always warmer than the 4 cm sensor.  This is different than the temperature patterns in Figure 10 and is confusing.  Are the colours in the legend correct, or are the colours of 4 and 14 cm mislabelled in the legend?

*We went through the data carefully. Some data files had been mixed up. It is all reordered now. We have also checked and modified the PANGAEA data files. Here is the new Figure 11.*

[Figure]

*Figure 11: Hourly time series of the Pt1000 thermistors from the SNOW3 station at heights of 4, 14, 29, and 64 cm.*

Section 6.4 - Do you intend for all dates shown in Figure 14 to be described/accounted for in the text? Please check the dates in the text description against the dates of the actual measurements, as it looks like 2012 is missing from the text.

*Thank you, sorry about that. As detailed in more depth in response to additional comments about the data files, we have modified the density and SSA data files and pit selection. We have entirely redrawn this Figure to also include data from the FOREST site. Here is the new Figure 14:*

[Figure]

*Figure 14: Vertical profiles of snow density and SSA measured in the vicinity of the TUNDRA station from 2012 to 2019 and near the FOREST station from 2016 to 2018.*

Figure 14 - The legend is covering over some of the data in both plots, but especially the Density plot.  Please move the legend so that all the data can be seen.

*This has been fixed. Please see the new Figure 14 above.*

Figure 16 - Should "volume" be removed after "(SWC)" in the Figure description text?

*We changed "soil water content (SWC) volume" to "soil volume water content (SWC)". This is now Figure 17 following the swapping of sections 7.1 and 7.2.*

Figure 18 - Please write dates with the same format as in Figure 14.

*We have moved the legend and harmonized the curves so that a given measurement date appears the same on both plots. Note that this is now Figure 15.*

[Figure]

*Figure 15: Thermal properties of soils. (a) Thermal conductivity; (b) density. Depths were measured from the base of the live lichen.*

Line 453 - "...which showed poor performance when simulating Arctic snowpack properties."  Perhaps add "have previously" before "showed" and provide a reference(s)?

*Done, thank you. We have added Domine et al., 2019 as a reference, line 484.*

Lines 466-467 - I assume these lines will be removed upon publication?  Along with "(dataset in review)" that is currently written twice in the header of each data file?

*Indeed, this has been done.*

**Data files:**

In the data files, what is the time zone of the time stamp?  Is it local time?  If so, which time zone is it?  This should be specified in each data file and/or in the manuscript.

*Thank you for this comment. The time zone in the original PANGAEA data files was UTC-4. However, PANGAEA requests that all data on the repository be UTC. We have therefore prepared new files and made a new submission to PANGAEA which corrects this, along with all the point raised below. On 7 December, PANGAEA informed us that due to a high submission rate, they cannot process our new files before 12 weeks. The new files can however be seen at:*

*https://www.dropbox.com/scl/fo/mjacraie1jf7dz0ly9mu3/h?rlkey=m5c2qcocskrqsafo3judgt0no&dl=0*

*In our new text, we specify lines 95 and 496 that times are UTC.*

Are the files tab-delimited?  The format of the files should be stated in the manuscript.

*Yes. This is specified in the PANGAEA front matter: "Download ZIP file containing all datasets as tab-delimited text". We have added this information in the Data availability statement, line 496.*

Please check the headers in the different data files to confirm that they are all consistent.

*All data files have been carefully checked throughout and corrections have been made where required.*

Umiujaq_rad_forest.tab and Umiujaq_rad_tundra.tab:  There are only 3 columns of data in some places, instead of 4 (for example, 2021-10-14T07:00 to 17:00, in both files).  Please check entire files for complete data presentation.

*Shortwave upwelling data was missing for the last day of the file only. This has been corrected.*

*https://www.dropbox.com/scl/fi/ibnskz1lhusvj570e4c75/Umiujaq_rad_forest_UTC.tab?rlkey=0 31243broc8xg44ddhcquuzbj&dl=0*

*https://www.dropbox.com/scl/fi/yb1965rp5iboybo2c0iyi/Umiujaq_rad_tundra_UTC.tab?rlkey =uga06gl93h00jx38ubk4i31g7&dl=0*

Why are there no quality flags in Umiujaq_rad_forest.tab (but they are there in Umiujaq_rad_tundra.tab)?

*We have added quality flags for all 4 radiation components. Since there was no power outages or instrument failures, all upwelling radiation data have QF=0. For downwelling radiation, QF=1 when there was frost on the sensors. This was detected from the value of the raw downwelling longwave radiation, also considering temperature (T<0 °C) and low wind speed values, as detailed for the TUNDRA radiation description. This will also be reminded for these data in the revision.*

Umiujaq_rad_tundra.tab:  QF SWD should not have a decimal point (should be just one digit).

*Thank you. This has been fixed.*

Umiujaq_precip_tundra.tab:  There are sections where there is rain (mm), but no precip (mm/hr) for many hours.  Is this possible?  For example, 2019-07-10T15:00 through 2019-07-11T19:00.  Can you explain the calculation method or the function of the field instrumentation which would allow this seeming inconsistency?

*Thank you for spotting this. We checked the file thoroughly. All the precipitation data are correct. The seasonal sums are correct as well. The snow-rain partitioning had errors starting on 28 May 2016, when we started using data from our Geonor gauge. The error was due to a shift when copying the data columns. This has now been fixed.*

*https://www.dropbox.com/scl/fi/az4kexkomxk4jsm95pz26/Umiujaq_precip_tundra_UTC.tab?r lkey=edel6k8j8bvembs9epu44ug3t&dl=0*

Umiujaq_snow_density.tab and Umiujaq_snow_surface_area.tab - what do locations A and B mean?  Please explain/briefly describe in the manuscript.

*This means pit A and pit B. We have modified the files and now label the column "Daily pit number" and have 1 or 2 instead of A or B. We hope this will be clear.*

*https://www.dropbox.com/scl/fi/h3aql2zoua86tc66cz3ai/Umiujaq_snow_density_UTC.tab?rlk ey=ixpuq79g23hwgzuk2t09dtyvk&dl=0*

https://www.dropbox.com/scl/fi/k4t1ywfws639p83u4vbz3/Umiujaq_snow_surface_area_UTC.tab?rlkey=bqfazo2bqu2bgrx2wzok0jv6t&dl=0

Please explain if snow surveys were not actually taken at midnight (all time stamps indicate 00:00). This also applies to Umiujaq_snow_temp_conduct.tab and Umiujaq_snow_temp_conduct_st2.tab files - time stamps are all 00:00.

*A snowpit study can take up to 4 hours for deep snow pits, so the time of day is not useful. For consistency with other files, we had just written T00:00. We have removed the time in the new files and just indicate the date, provided that this is allowed by PANGAEA. If not, we will add noon as the time.*

Umiujaq_snow_density.tab and Umiujaq_snow_surface_area.tab: Lat/long are missing from many of the 2018 FOREST density and surface area measurements. Please explain.

*We have modified the file to indicate the position. In fact, while we were at it, we also slightly modified our pit selection. Our original criterion was based primarily on the proximity to the met stations. Now we rather selected pits with more similar vegetation cover. We have also therefore modified Figure 14 accordingly and also added 2 panels to represent graphs for the FOREST site as well, as detailed in a response to a previous comment. Note that for clarity, Figure 14 does not show all the pits present in the data file, but a representative selection. We have also added a few details on snowpit work, lines 398-414.*

*Umiujaq_snow_height.tab:* "heght" in the header (above the "Keyword(s)" line) needs an "I"

*This has been modified.*

https://www.dropbox.com/scl/fi/vsut8qkqm9zatinlbkokk/Umiujaq_snow_height_UTC.tab?rlkey=z6fukcsrpdcufwtotrl9gvdb5&dl=0

Umiujaq_snow_surface_area.tab: "differnt" (just above "Keyword(s)") needs an "e"

*This has been modified.*

Umiujaq_snow_temp_conduct_st2.tab: Temperatures are missing from some intervals, including 26 April 2019, 6 May 2019, and others. Please explain.

*The file has been carefully checked and some missing data has been filled. However, we now explain in the text of the revised version, line 382, that there is a temperature threshold for the thermal conductivity measurements, which generates data gaps if the snow is too warm. The threshold is because a thermal conductivity measurement heats up the snow by about*

*2°C. If the snow is too warm, there is a risk of melting and of irreversible modification of the snow structure, which must be avoided. Therefore, if the snow is warmer than -2.5°C, then no measurement is made.*

*[https://www.dropbox.com/scl/fi/07ivme3gpict87dnvs69h/Umiujaq_snow_temp_conduct_st2_UTC.tab?rlkey=vzo1r8y78oxvck0tm86iefwjy&dl=0](https://www.dropbox.com/scl/fi/07ivme3gpict87dnvs69h/Umiujaq_snow_temp_conduct_st2_UTC.tab?rlkey=vzo1r8y78oxvck0tm86iefwjy&dl=0)*

*Please see also response to next comment.*

Umiujaq_snow_temp_conduct.tab:  Temperatures are missing in some intervals, conductivity is missing in others, and both are missing in a fair number of intervals.  Why include these rows, if there are no measurement data at those times?  Is it because it is summer and there is no snow?  Why is this file made differently than Umiujaq_snow_temp_conduct_st2.tab, which seems to have removed all the summer months with no snow?

*The file has been completely rebuilt. Moreover, for the TUNDRA SNOW1 post, we have added 3 years of data coming from the first post that was placed in February 2013. Some data gaps remain, which indicate that the measurement could not be made. There was also a power outage in February 2014 with missing data. We chose not to remove these data gaps or missing occasional values so that the file will show a continuous time series, rather than erratic-looking dates. We have however removed all the summer values. For the FOREST SNOW3 post, we have also added some data at 4 cm height. Because of the thick snow cover, temperature was often >-2.5°C, so that no measurement was made at all during the whole 2017-2018 and 2020-2021 winters. However, the measurements that were performed during the 2018-2019 and 2019-2020 winters are now included in the data file.*

*[https://www.dropbox.com/scl/fi/issaq4e8pj6uyrs3t32ld/Umiujaq_snow_temp_conduct_UTC.tab?rlkey=quftlgdqc27at1y47arzau6iw&dl=0](https://www.dropbox.com/scl/fi/issaq4e8pj6uyrs3t32ld/Umiujaq_snow_temp_conduct_UTC.tab?rlkey=quftlgdqc27at1y47arzau6iw&dl=0)*

Umiujaq_temp_wind_forest.tab:  Why is there no quality flag for the Temp in temp_wind_forest, though there is one in Umiujaq_temp_wind_tundra.tab?

*At TUNDRA, there were temperature data gaps that required gap filling, some of them with a similar instrument located on the nearby tower. At FOREST, as indicated lines 140-141, there were no data gaps, besides a few gaps <3h that were filled by interpolation. Therefore, all data have the same quality and we thought there was no need for a quality flag. We have nevertheless modified the file to add QF=0 to all temperature data points.*

*https://www.dropbox.com/scl/fi/jpwce80m2lgtvfrbtg7if/Umiujaq_temp_wind_forest_UTC.tab?rlkey=dwogdzu6ndtwjnjdu50zxqduf&dl=0*

---

## Author Response (AR2)

Dear Kirsten,

Thank you for accepting our paper. The PANGAEA DOI for our data set is: https://doi.pangaea.de/10.1594/PANGAEA.964743 . This has been included in the paper.

Best regards,

Florent Domine, on behalf of all coauthors.